# Examining the perceived impact of the COVID-19 pandemic on cervical cancer screening practices among clinicians practicing in Federally Qualified Health Centers: A mixed methods study

**Lindsay Fuzzell[1]\*[†], Paige Lake[1]\*[†], Naomi C Brownstein[2], Holly B Fontenot[3], Ashley Whitmer[1], Alexandra Michel[3], McKenzie McIntyre[1], Sarah L Rossi[4], Sidika Kajtezovic[4], Susan T Vadaparampil[1,5], Rebecca Perkins[4]**

[1]H. Lee Moffitt Cancer Center & Research Institute, Health Outcomes and Behavior, Tampa, United States; [2]Medical University of South Carolina, Public Health Sciences, Charleston, United States; [3]University of Hawaii at Manoa, Nancy Atmospera-Walch School of Nursing, Honolulu, United States; [4]Boston University, Chobanian & Avedisian School of Medicine, Boston, United States; [5]H. Lee Moffitt Cancer Center & Research Institute, Office of Community Outreach, Engagement, and Equity, Tampa, United States

**\*For correspondence:**
Lindsay.Fuzzell@moffitt.org (LF); paige.lake@moffitt.org (PL)

[†]These authors contributed equally to this work

**Competing interest:** The authors declare that no competing interests exist.

## Abstract

**Background:** The COVID-19 pandemic led to reductions in cervical cancer screening and colposcopy. Therefore, in this mixed methods study we explored perceived pandemic-related practice changes to cervical cancer screenings in federally qualified health centers (FQHCs).

**Methods:** Between October 2021 and June 2022, we conducted a national web survey of clinicians (physicians and advanced practice providers) who performed cervical cancer screening in FQHCs in the United States during the post-acute phase of the COVID-19 pandemic, along with a sub-set of qualitative interviews via video conference, to examine perceived changes in cervical cancer screening practices during the pandemic.

**Results:** A total of 148 clinicians completed surveys; a subset (n=13) completed qualitative interviews. Most (86%) reported reduced cervical cancer screening early in the pandemic, and 28% reported continued reduction in services at the time of survey completion (October 2021- July 2022). Nearly half (45%) reported staff shortages impacting their ability to screen or track patients. Compared to clinicians in Obstetrics/Gynecology/Women's health, those in family medicine and other specialties more often reported reduced screening compared to pre-pandemic. Most (92%) felt that screening using HPV self-sampling would be very or somewhat helpful to address screening backlogs. Qualitative interviews highlighted the impacts of staff shortages and strategies for improvement.

**Conclusions:** Findings highlight that in late 2021 and early 2022, many clinicians in FQHCs reported reduced cervical cancer screening and of pandemic-related staffing shortages impacting screening and follow-up. If not addressed, reduced screenings among underserved populations could worsen cervical cancer disparities in the future.

**Funding:** This study was funded by the American Cancer Society, who had no role in the study's design, conduct, or reporting.

## Editor's evaluation

This US study presents findings from an online survey and in-person interviews of healthcare providers in areas associated with cervical screening provision during the post-acute phase of the pandemic. The findings are valuable as they provide insights into a range of areas, from healthcare characteristics to screening barriers and HPV self-sampling. The evidence supporting the claims of the authors is solid. The work will be of interest to public health scientists and a cancer prevention and control audience.

## Introduction

Cervical cancer prevention via screening and treatment of pre-invasive disease has dramatically reduced cervical cancer incidence and mortality rates (*Sawaya and Huchko, 2017*). However, lack of access to screening and treatment services results in geographic, racial/ethnic, and socioeconomic disparities in cervical cancer incidence and mortality (*Buskwofie et al., 2020*; *Vu et al., 2018*; *Chen et al., 2012*; *Akers et al., 2007*). A recent study of cervical cancer patients showed that over half were either never screened or were overdue for screening (*Benard et al., 2021*). Lack of screening remains the most common reason why individuals develop cervical cancer in the United States (US) and worldwide. In the US, cervical cancer screening is considered a critical element of preventive healthcare, and the addition of Human Papillomavirus (HPV) testing, along with Pap testing, can improve prevention programs by allowing longer screening intervals for patients testing negative, while providing more precise risk estimates to allow evidence-based management of patients with abnormal screening results (*Schiffman et al., 2011*; *Leinonen et al., 2009*; *Mayrand et al., 2007*).

Since the COVID-19 pandemic began in the US in 2020, however, cancer screenings decreased for many cancer types (*Chen et al., 2021*; *Poljak et al., 2021*; *Amram et al., 2022*; *Smith and Perkins, 2022*), with cervical cancer screening decreasing more than others (*Miller et al., 2021*; *Mayo et al., 2021*; *Fedewa et al., 2022*). Early in the pandemic, patient fear of contracting COVID-19 and reduction in non-urgent medical services impacted the ability to perform cervical cancer screening and colposcopy (*Massad, 2022*). Federally qualified health centers (FQHCs) in the US are government funded health centers or clinics that provide care to medically underserved populations. Maintaining cancer screening in these and other safety net facilities is critical as they serve patients at the highest risk for cervical cancer: publicly insured/uninsured, immigrant, and historically marginalized populations (*Adams et al., 2020*; *Fisher-Borne et al., 2021*). A survey of 22 federally qualified health centers (FQHCs) that conducted cervical cancer screenings in 2020 found that 90% reported cancelling cervical cancer screenings during the height of the pandemic (*Fisher-Borne et al., 2021*). While 86% reported rescheduling cancer screenings for future visits, the success of this strategy to maintain screening rates was not measured. FQHCs reported strategies such as switching to telehealth visits and implementing in-office structural changes, new waiting room protocols, and new referral processes to address pandemic restrictions (*Fisher-Borne et al., 2021*). Following widespread vaccination and the resumption of in person services, cancer screening rates have begun to rebound (*Chen et al., 2021*; *McBain et al., 2021*), but challenges still exist. Currently, medical staff shortages and backlogs of patients needing to catch up on preventive services lead to longer wait times for scheduling appointments and decreased screening rates (*Smith and Perkins, 2022*; *Massad, 2022*; *Wentzensen et al., 2021*).

Little work has explored the impact of the COVID-19 pandemic on clinician perceptions of cervical cancer screening and staffing challenges in FQHCs. In order to identify characteristics that could be targets for future interventions or additional supports, this paper examines the association of clinician characteristics with perceived changes in cervical cancer screening and the impact of pandemic-related staffing changes on screening and abnormal results follow-up during the pandemic period of October 2021 through July 2022 in FQHCs and safety net settings of care.

## Methods

### Participant recruitment and target population

The target population were clinicians, defined for the purpose of this study as physicians and Advanced Practice Providers (APPs), who conducted cervical cancer screening in federally qualified health centers and safety net settings of care (hereafter referred to as 'FQHCs') in the United States during the post-acute phase of the COVID-19 pandemic. Clinicians were eligible to participate if they: (1) performed cervical cancer screening, (2) were a physician or APP, and (3) were currently practicing in an FQHC in the US between October 2021 and July 2022, the post-acute period of the pandemic in the US when COVID-19 vaccination was widely available to the general population. We recruited clinicians for participation in the online survey hosted via Qualtrics through periodic recruitment email messages via the American Cancer Society Vaccinating Adolescents Against Cancer (VACs) program and the professional networks of the PIs (RBP, STV).

Survey participants were asked if they would also be willing to participate in qualitative interviews via phone. A random sample of those who indicated willingness were contacted for participation. This study was approved by Moffitt Cancer Center's Scientific Review Committee and Institutional Review Board (MCC #20048) and Boston University Medical Center's Institutional Review Board (H-41533). All survey participants viewed an information sheet in lieu of reading and signing an informed consent form, and interview participants provided verbal consent. All were compensated for their time completing the survey or interview.

### Survey development and validation

Quantitative survey questions were developed based on recent literature exploring the effects of the COVID-19 pandemic on cancer screening practices (*Miller et al., 2021*; *Wentzensen et al., 2021*) and the investigators' clinical observations. The draft survey was reviewed by an expert panel of FQHC providers (n=8), refined, piloted, and finalized after incorporating pilot feedback and testing technical functionality of the Qualtrics survey among the study team.

Clinician characteristics assessed included age, race/ethnicity, training, specialty, and geographic region. Age was measured in years and categorized for analysis as <30, 30–39, 40–49, 50+. Gender identity was assessed as male, female, transgender, and other. Race was assessed as Asian, Black/African American, White, Mixed race, Native Hawaiian/Pacific Islander, American Indian/Alaska Native, and Other. Ethnicity was assessed as Hispanic/Latinx or non-Hispanic/Latinx. Race/ethnicity was categorized for analysis as White non-Hispanic versus all others due to small cell sizes of non-White and Hispanic participants. For all variables assessed in this manuscript that allowed write-in/free responses, responses were re-classified within the pre-determined categories for each variable when possible.

Clinician training was assessed as physician (medical doctor [MD], doctor of osteopathic medicine [DO]) or advanced practice providers (APPs) (physician assistant [PA], nurse practitioner [NP], and certified nurse midwife [CNM]). Clinician training was categorized as: (1) MD/DO (doctors of medicine and osteopathic medicine) and (2) APPs (NPs, CNMs, PAs). Clinical specialty was assessed as Obstetrics and Gynecology (OBGYN), family medicine, internal medicine (IM), pediatric/adolescent medicine, women's health, and other (via write in). Based on prior literarature (*Neugut et al., 2019*) and the number of respondents in each category, we created the following categories for clinician specialty: (1) Women's Health/OBGYN, (2) Family Medicine, and (3) IM, Pediatrics/Adolescent Medicine. Geographic location included four US regions (Northeast, South, Midwest, West) and a non-responder category for those who did not provide state or zip code. Based on national data indicating geographic variation in coverage rates by US region (*Buskwofie et al., 2020*) as well and distribution of respondents, region was categorized as (1) Northeast, (2) South, and (3) West and Midwest.

We also assessed clinical behaviors and attitudes associated with cervical cancer screening. Questions captured the number of screens performed monthly, test(s) used for screening, attitudes toward using self-collected HPV testing for cervical cancer screening, barriers to screening, tracking systems, and staffing changes.

Qualitative interview guide questions were developed based on recent literature (*Miller et al., 2021*; *Wentzensen et al., 2021*) and the investigators' clinical observations. The draft interview guide was reviewed by an expert panel of FQHC providers (n=8) and revised. Interview questions explored survey topics in depth, including experiences with providing cervical cancer screening at different

points during the pandemic, barriers to providing care, as well as strategies for improving follow-up, including tracking systems and self-sampled HPV testing.

## Data analysis

### Quantitative survey data

We assessed descriptive statistics (frequencies, percentages) of clinician characteristics and outcome variables. We conducted separate exact binary logistic regressions (due to small cell sizes) examining the associations of clinician characteristics with (a) screening practices at the time of survey participation (the same/more versus less than pre-pandemic), and (b) pandemic-associated staffing changes impacting the ability to screen or follow-up (yes/no). The following variables were included in the full models for each outcome: race/ethnicity, age, gender, region, clinician training., clinician specialty. We used manual forward selection with a value for entry and significance of 0.10 because this strikes a balance between the commonly accepted method of using AIC (which assumes significance level of 0.157), and the often used alpha of 0.05, which could lead to failure to identify associations due to small sample size. Variables were added sequentially with the variable with the lowest p-value below 0.10 added at each step. We produced forest plots displaying odds ratios and confidence intervals from this model (Figure 2). Analyses were conducted in SAS version 9.4.

### Qualitative interview data

Interviews were conducted by three co-authors (RBP, AM, HBF) trained in qualitative methodology via video conference (Zoom); interviews were audio recorded and transcribed verbatim. Data were coded using thematic content analysis (*Elo and Kyngäs, 2008*). Based on the questions in the initial interview guide, a priori codes were developed and a codebook created to operationalize and define each code. The qualitative analysis team independently reviewed the data twice. The team hand coded the data with the initial codes and made notes on possible new codes in the first coding pass. Then, notes on possible new codes were discussed until consensus was reached. The codes were then revised and transcripts reviewed using the updated code categories. This second coding pass served to clean coding from the first coding pass and identify emergent themes not initially identified (*Unknown, 1998*). At least two coders reviewed each transcript. Discrepancies were resolved by discussion in weekly group meetings. A centralized shared data sheet was used for coding to facilitate communication across different institutions.

### Role of the funding source

This study was funded by the American Cancer Society, who had no role in the study's design, conduct, or reporting.

## Results

### Quantitative survey data

A total of 159 potential participants viewed the online study information sheet and completed screening items; 11 were excluded due to ineligible clinical training (n=5) or not conducting cervical cancer screening (n=6). Data were cleaned and invalid surveys were removed. Invalid surveys included potential duplicate responses identified by repeat IP address, nonsensical write-in free responses, and those with numerous skipped items. *Table 1* details clinician characteristics and screening practices of the final analytic sample (n=148). *Figure 1* provides a flow diagram describing the process of determining the final analytic sample size. The sample was primarily female (85%), White (70%), non-Hispanic (86%), and practiced in the Northeast (63%). Most (70%) reported specializing in family medicine, 19% reported Women's Health/OBGYN, and 11% reported other specialties. All but one participant (99%) used Pap/HPV co-testing for routine screening of patients aged 30–65, and 61% performed 10 or fewer screens per month. Most (93%) clinicians determined the next step in management themselves when their patients had abnormal results (rather than refer to a specialist). Most (78%) had colposcopy available on site, though only 31% of participants reported that treatment (Loop Electrosurgical Excision Procedure, [LEEP]) was available on site.

Most (95%) reported decreased screening during 2020 compared to pre-pandemic, and 53% stated that screening services were completely suspended at some point during the pandemic.

**Table 1.** Clinician characteristics and screening practices.

| Variable | Frequency | % | Valid N |
|---|---|---|---|
| **Clinician characteristics** | | | |
| **Age** | | | 147 |
| Less than 30 | 20 | 14 | |
| 30–39 | 56 | 38 | |
| 40–49 | 36 | 24 | |
| 50+ | 35 | 24 | |
| **Gender identity** | | | |
| Female* | 125 | 85 | 148 |
| Male | 22 | 15 | |
| Transgender/gender non-binary | 1 | 0.67 | |
| **Race** | | | 148 |
| Asian | 13 | 9 | |
| Black/African American | 15 | 10 | |
| Mixed race | 10 | 7 | |
| Other | 7 | 5 | |
| White | 103 | 70 | |
| **Ethnicity** | | | 148 |
| Hispanic/Latinx | 21 | 14 | |
| Not Hispanic/Latinx | 127 | 86 | |
| **Clinician Training** | | | 148 |
| MD and DO | 67 | 45 | |
| APPs[†] | 81 | 55 | |
| **Clinician Specialty** | | | 148 |
| Women's Health and Ob/GYN | 28 | 19 | |
| Family Medicine | 103 | 70 | |
| Internal Medicine, Pediatric/Adolescent Medicine, and 'other' | 17 | 11 | |
| **Region** | | | 148 |
| Northeast | 93 | 63 | |
| South | 28 | 19 | |
| West & Midwest | 26 | 18 | |
| Non-responders | 1 | 0.7 | |
| **Current number of cervical cancer screenings performed per month** | | | |
| 1–10 | 90 | 61 | |
| 11–20 | 27 | 18 | |
| >20 | 31 | 21 | |
| Pap/HPV co-testing as screening method for patients aged 30–65 | 147 [‡] | 99 | 148 |
| Respondent determines management following abnormal results (yes) | 138 | 93 | 148 |
| Health center provides colposcopy on site (yes) | 115 | 78 | 148 |
| Health center provides treatment (LEEP) on site (yes) | 46 | 31 | 148 |

*Table 1 continued on next page*

*Table 1 continued*

| Variable | Frequency | % | Valid N |
|---|---|---|---|
| PANDEMIC IMPACT ON SCREENING AND MANAGEMENT | | | |
| Screening in 2020 compared to pre-pandemic (less) [§] | 127 | 95 | 134 |
| Screening services stopped at any time during the pandemic (yes) [§] | 66 | 53 | 125 |
| Colposcopy services stopped at any time during the pandemic (yes) [§], [¶] | 36 | 31 | 115 |
| LEEP services stopped at any time during the pandemic (yes) [§], [¶] | 8 | 17 | 46 |
| Screening in 2021/now compared to pre-pandemic [§] | | | 140 |
| Less | 39 | 28 | |
| Same | 65 | 46 | |
| More | 36 | 26 | |

*for all percentages included in all tables, when percentages were .6-.9, we rounded up to the next whole number.

*Due to small numbers, transgender/non-binary/other were unable to be analyzed as their own category. They were assigned to female for regression analyses because female was the most common response. No difference was noted when grouped with male.

[†]APPs included: NPs (52), CNMs (7), PAs (17), and other (5).

[‡]The remaining respondent used primary HPV testing. No respondents in this sample used cytology alone.

[§]Participants who selected 'unsure' were excluded from the denominator. 14 (9%) participants were unsure whether screening was less in 2020 compared to pre-pandemic, 23 (16%) were unsure whether screening services were stopped at any time, 53 participants (36%) were unsure whether colposcopy practices were stopped, 21 (14%) were unsure whether LEEP services were stopped, and 8 (5%) were unsure whether they were screening more or less in 2021/now compared to pre-pandemic.

[¶]Participants who did not indicate that they performed colposcopy and LEEP services on site were excluded from the demonimator.

Smaller proportions reported suspensions of colposcopy (31%) and LEEP (17%) services. By October 2021-July 2022, when the survey was conducted, screening had recovered somewhat. Approximately one-quarter (28%) reported less cervical cancer screening currently than before the pandemic, 46% reported the same amount, and 26% more screening. Among clinics providing LEEP services, 76% had currently resumed pre-pandemic LEEP capacity (data not shown).

We examined cervical cancer screenings performed monthly by clinician training and specialty (*Table 2*). Overall, 32% of clinicians screened 1–5 patients monthly, 29% screened 6–10 patients, 18% screened 11–20 patients, and 21% reported screening >20 patients. Approximately 18% of MD/DOs and 23% of APPs screened >20 patients per month, while 37% of MD/DOs and 27% of APPs screened 1–5 patients per month. Screening practices varied by specialty, with 59% of clinicians in OBGYN/Women's Health screening >20 patients per month compared to 11% in Family Medicine.

*Table 3* and *Figure 2* detail logistic regression model results for clinician and practice characteristics associated with odds of doing the same amount or more cervical cancer screening at the time of survey completion (2021–2022) as compared to before the COVID-19 pandemic. Region, gender, and age were not included in the model after completing the specified variable selection process. Clinician specialty was significantly associated with odds of doing the same or more cervical cancer screening at time of the survey (2021–2022) than before the pandemic (p=0.04). Compared to Women's Health/OBGYNs, those who identified as family medicine clinicians and other were significantly associated with decreased odds of performing the same or more screening at time of survey (2021–22) (Family medicine: OR = 0.29, *95% CI: 0.08–1.07, p=0.06; Other: OR = 0.12, 95% CI: 0.025–0.606, p=0.01*). Further, clinician training was significantly associated with increased odds of doing the same or more screening at time of the survey (2021–2022) as compared to before the pandemic (p = 0.06); compared to MDs/DOs, APPs had higher odds of performing the same or more screening at time of the survey (2021–2022) *(OR = 2.15, 95% CI: 0.967–4.80, p=0.06)*. Clinician race/ethnicity was also significantly associated, with non-White clinicians more likely to report the same or more screening at time of the survey (2021–2022) as compared to White non-Hispanic clinicians *(OR = 2.16, 95% CI:.894–5.21, p=0.08)*.

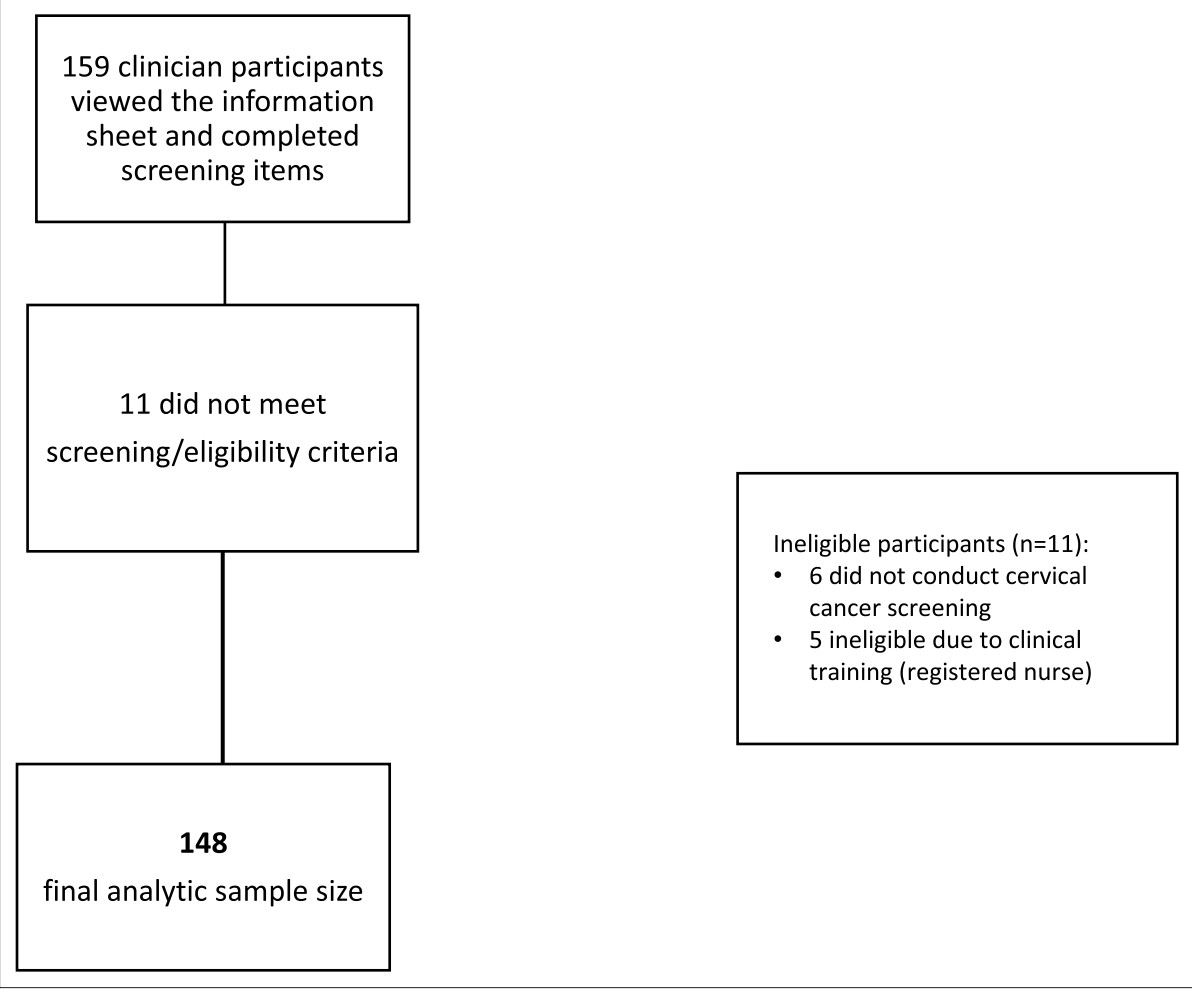

**Figure 1.** Study flow chart depicting participant exclusions and final analytic sample.

Clinicians reported various barriers to cervical cancer screening (*Table 4*). The following were 'often' considered barriers by respondents: limited in-person appointment availability (45%), patients not scheduling (57%) or attending appointments (42%), switching to telemedicine (33%) and the need to address more pressing health concerns (31%). Another important barrier was pandemic-associated

**Table 2.** Cervical cancer screenings performed monthly by clinician specialty and clinician training.

| | 1–5 patients per month N=47 | 6–10 patients per month N=43 | 11–20 patients per month N=27 | >20 patients per month N=31 | Total N=148 |
|---|---|---|---|---|---|
| **Clinician Training** | | | | | |
| MD/DO | 25 (37%) | 20 (30%) | 10 (15%) | 12 (18%) | 67 |
| APPs | 22 (27%) | 23 (28%) | 17 (21%) | 19 (23%) | 81 |
| **Clinician Specialty** | | | | | |
| OBGYN/Women's Health | 2 (7%) | 4 (14%) | 6 (21%) | 17 (59%) | 29 |
| Family Medicine | 39 (38%) | 34 (33%) | 19 (18%) | 11 (11%) | 103 |
| IM, Peds/Adol. Med. | 6 (38%) | 5 (31%) | 2 (13%) | 3 (19%) | 16 |

Placeholder for **Figure 1**\*Study flow chart depicting participant exclusions and final analytic sample.

**Table 3.** Final model of clinician and practice characteristics associated with odds of reporting conducting the same amount or more cervical cancer screening now/in 2021 than before the COVID-19 pandemic (N=140).

Manual forwards selection was utilized and the following variables were not selected for the final model (p>0.10): (1) region (2) gender and (3) age.

| | Overall p | B | SE | Adjusted odds ratio | p | CI |
|---|---|---|---|---|---|---|
| Clinician training | 0.0605 | | | | | |
| APPs | | 0.7676 | 0.4089 | 2.155 | **0.0605** | 0.967–4.802 |
| MD/DO | | - | - | - | - | - |
| Clinician specialty | 0.0364 | | | | | |
| Family Medicine | | −1.2214 | 0.6594 | 0.295 | **0.0640** | 0.081–1.07 |
| Int. Med., Peds/Adol. Med. | | −2.0996 | 0.8159 | 0.123 | **0.0101** | 0.025-.606 |
| Women's Health/OBGYN | | - | - | - | - | - |
| Clinician race/ethnicity | 0.0873 | | | | | |
| All other races/ethnicities | | 0.7694 | 0.4500 | 2.1159 | **0.0873** | 0.894–5.214 |
| White non-Hispanic | | - | - | - | - | - |

*CI reported is for OR.

*Placeholder for **Figure 2*** Forest plots depicting clinician and practice characteristics associated with odds of reporting conducting the same or more cervical cancer screening now/in 2021 vs. before the pandemic.

staffing changes impacting the ability to screen for cervical cancer, track abnormal results, or follow-up with their patients, which was reported by 45% of participants. Approximately half of participants reported current decreased staffing levels of medical assistants (56%), and office staff (43%) as compared to pre-pandemic while approximately one third reported decreases in physicians (35%), APPs (28%), and nurses (28%). Only 12% reported lack of health insurance was an important barrier to screening.

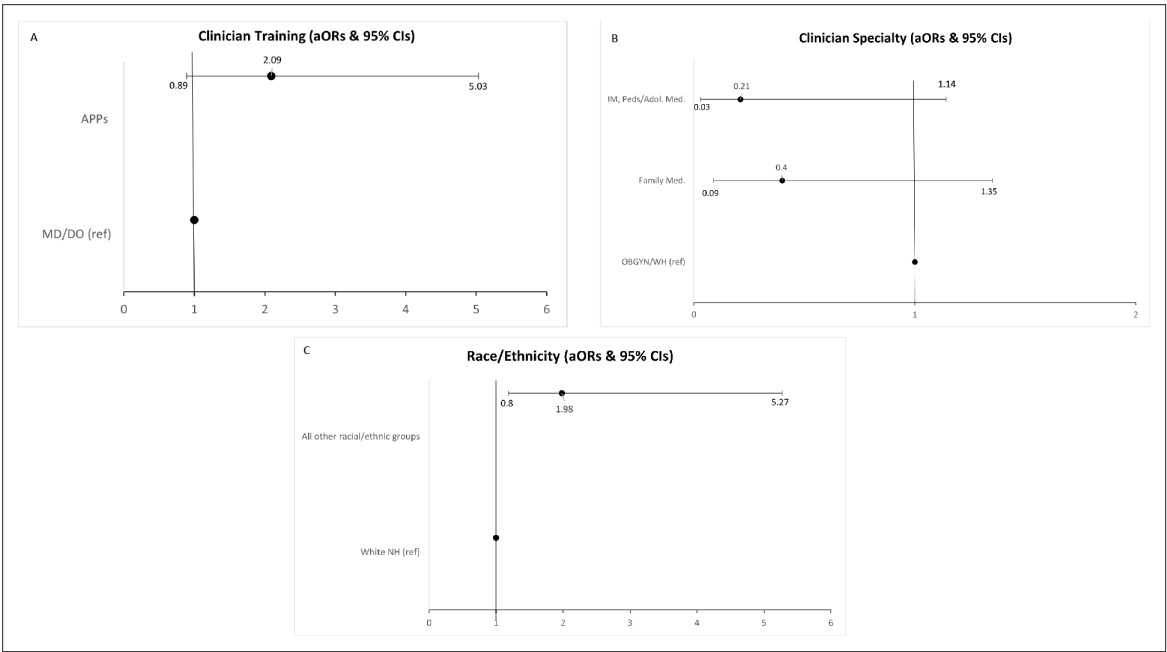

**Figure 2.** Forest plots depicting clinician & practice characteristics associated with odds of reporting conducting the same amount or more cervical cancer screening now/in 2021 vs before the pandemic.

**Table 4.** Barriers to cervical cancer screening and strategies for tracking patients.

| BARRIERS | Rarely n (%) | Sometimes n (%) | Often n (%) | Unsure n (%) | Valid N |
|---|---|---|---|---|---|
| **Systems barriers** | | | | | 148 |
| Limited in-person appointment availability at our health center | 24 (16) | 53 (36) | 66 (45) | 5 (3) | |
| Patients not scheduling appointments | 5 (3) | 50 (34) | 85 (57) | 8 (6) | |
| Patients not attending appointments (no shows) | 8 (6) | 73 (49) | 62 (42) | 5 (3) | |
| Patient lack of health insurance or limited coverage* | 83 (56) | 36 (24) | 18 (12) | 11 (8) | |
| Inability to track patients who are due for screening | 58 (39) | 46 (31) | 32 (22) | 12 (8) | |
| Health center (or providers) not prioritizing screening due to need to address more acute health problems | 34 (23) | 61 (41) | 46 (31) | 7 (5) | |
| Switched to telemedicine visits so screening not available | 34 (23) | 59 (40) | 48 (33) | 6 (4) | |
| **Staffing barriers** | Frequency | | Percent | | 148 |
| COVID-related staffing changes impacted ability to screen or track abnormal results (yes) | 67 | | 45 | | |
| Current health center staffing compared to pre-pandemic | Decreased n (%) | Stayed the same n (%) | Increased n (%) | Unsure n (%) | 148 |
| Physician (MD, DO) | 52 (35) | 80 (54) | 6 (4) | 10 (7) | |
| Nurse practitioner, Physician Assistant, Certified Nurse Midwife, other Advanced Practice Provider | 42 (28) | 71 (48) | 22 (15) | 13 (9) | |
| Nurse (RN, LPN) | 42 (28) | 71 (48) | 22 (15) | 13 (9) | |
| Medical Assistant | 83 (56) | 45 (30) | 8 (6) | 12 (8) | |
| Office Staff | 64 (43) | 64 (43) | 6 (4) | 14 (10) | |

*Participants were also asked what proportion of patients were unable to obtain treatment (LEEP) due to financial issues, 70% (n=102) answered 0–20%.

Clinician and practice characteristics associated with odds of reporting staff shortages, tracking abnormal results, and follow-up were also assessed using logistic regression. In manual forwards selection, gender, region, age, race/ethnicity, clinician specialty and training were not selected for the final model, indicating no factors significantly associated with staffing shortages. *Table 5* highlights results related to strategies for tracking patient screening and abnormal results. To address missed care during the pandemic, most participants reported scheduling screening at the time of telemedicine visits (74%), performing screening when patients presented for other concerns (61%), and querying electronic medical records (62%). Few (22%) reported extra clinical sessions or extended hours devoted to screening. A minority (20%) reported that they did not have any system to track patients overdue for screening. The most commonly reported tracking systems for screening included the electronic medical record (63%) and dedicated staff members (25%). When asked about management of abnormal screening test results, participants most commonly reported that they were not aware of a tracking system (38%). When systems were in place, they included: electronic medical record tracking (34%), a dedicated staff member (36%), and paper logs (5%).

HPV self-sampling has been proposed as a method to improve cervical cancer screening rates. *Table 6* highlights clinician attitudes towards adopting HPV self-sampling as a strategy. A total of 31% felt that self-sampling would be very helpful and 61% felt it would be somewhat helpful to address pandemic-associated screening deficits. Approximately half (49%) would offer self-sampling only to patients who were unable to complete in-clinic screening, 35% would offer to any patient who preferred to self-sample, 6% would enact self-sampling for all patients, and 5% would not offer self-sampling. The most common perceived benefits of self-sampling were screening patients who had difficulty undergoing speculum exams (26% moderate benefit, 56% large benefit), or screening patients who had access to care issues (34% moderate benefit, 39% large benefit). However, clinicians reported concerns about patients collecting inadequate samples (33% moderate, 33% large concern), not returning specimens in a timely manner (35% moderate, 38% large concern), or not presenting

**Table 5.** Strategies for tracking patients and catching up on missed screenings*.

| STRATEGIES | Frequency | Percent | Valid N |
|---|---|---|---|
| Policies or plans for catching up on screenings that were missed due to the pandemic | | | 148 |
| Patients seen via telemedicine are scheduled for future screening visits | 110 | 74 | |
| Electronic medical record is queried to identify patients who are overdue | 92 | 62 | |
| Added dedicated cervical cancer screening days/hours | 32 | 22 | |
| Try to perform cervical cancer screening at acute problem visits/take advantage of opportunities to screen during unrelated visits | 90 | 61 | |
| System for tracking patients overdue for screening | | | 148 |
| No, unaware of any system | 29 | 20 | |
| Paper log of patients | 5 | 3 | |
| Each dept. has its own system | 5 | 3 | |
| Electronic medical record tracker | 94 | 63 | |
| Dedicated staff person/team member to review records and contact patients | 37 | 25 | |
| Other | 16 | 11 | |
| System for tracking abnormal results (e.g., colposcopy referrals) | | | 148 |
| Paper log of patients | 8 | 5 | |
| Each dept. has its own system | 7 | 5 | |
| I am not aware of any system/each provider tracks own results | 56 | 38 | |
| Electronic medical record tracker | 50 | 34 | |
| Dedicated staff person to review records and contact patients | 53 | 36 | |
| Other | 16 | 11 | |

Note, participants were asked to check all that apply therefore answers sum to >100%.

for other primary care services (33% moderate, 31% large concern). Participants were able to add free text to explain their answers in this section. Several participants who expressed concerns about HPV self-sampling described negative experiences with poor return rates and inadequate samples in home-based colon cancer screening.

## Qualitative data

A total of 15 clinicians participated in qualitative interviews. The qualitative sub-sample was primarily female (93%), White (67%), non-Hispanic (100%), and practiced in the Northeast (67%). More than half (53%) were APPs, and 73% specialized in Family Medicine. Three themes emerged in the qualitative analysis including: initial pandemic-associated barriers, ongoing barriers (systems and staffing), facilitators and strategies for catching up on cervical cancer screening (*Table 7*).

## Initial pandemic-associated barriers

These initial barriers were related to closing of offices/limiting office visits, patient fear of in-person care, prioritizing acute/urgent health conditions over preventive care, and inability to provide cervical cancer screening during telemedicine visits. In primary care offices, early disruptions were associated with caring for persons with COVID-19: "People working, especially in family medicine, were distributed to the COVID clinic… And so non-essential visits including routine pap smears were put on hold" (APP, Family Medicine). Many clinics switched to telemedicine, which was helpful to address acute issues, but reduced opportunities for cervical cancer screening. One said: "If they had been in the clinic… I would have probably done cervical cancer screening at that time." This participant noted

**Table 6.** HPV self-sampling perceptions and practices.

| | Frequency | % | Valid N |
|---|---|---|---|
| **Helpfulness of HPV self-sampling to catch up patients overdue for screening due to COVID-19 pandemic** | | | 147 |
| Not helpful | 12 | 8 | |
| Somewhat helpful | 89 | 61 | |
| Very helpful | 46 | 31 | |
| **Would recommend HPV self-sampling instead of clinician-collected sample for cervical cancer screening** | | | 148 |
| All patients | 9 | 6 | |
| Any patient who preferred a self-sample over a clinician-collected sample | 52 | 35 | |
| Only pts. who couldn't have screening in clinic because of transportation issues, fear of coming to clinic, difficulty with speculum exams | 72 | 49 | |
| N/A I would not offer HPV self-sampling | 8 | 5 | |
| Other | 7 | 5 | |
| **Location to perform self-sample HPV tests** | | | 148 |
| In clinic | 8 | 6 | |
| At home | 9 | 6 | |
| Either in clinic or home, depending on pt. preference | 120 | 86 | |
| Other | 3 | 2 | |

| **Benefits/advantages of self-sampled HPV testing** | Not a benefit n (%) | Small benefit n (%) | Moderate benefit n (%) | Large benefit n (%) | **147** |
|---|---|---|---|---|---|
| Screen patients who have difficulty accessing screening due to lack of qualified providers, distance to clinic, or logistical barriers (e.g., childcare or work schedules) | 7 (5) | 32 (22) | 50 (34) | 58 (39) | |
| Screen patients via telemedicine | 10 (7) | 50 (34) | 44 (30) | 43 (29) | |
| Screen patients who would prefer not to have speculum exams (e.g. mobility issues or history of trauma) | 3 (2) | 23 (16) | 38 (26) | 83 (56) | |
| **Concerns about self-sampled HPV testing** | Not a concern | Small concern | Moder-ate concern | Large concern | 147 |
| A pelvic exam by a clinician should be part of cervical cancer screening | 20 (13) | 57 (39) | 38 (26) | 32 (22) | |
| Patients may not collect adequate specimens | 4 (3) | 45 (31) | 49 (33) | 49 (33) | |
| Patient may not return specimen in a timely manner | 3 (2) | 37 (25) | 51 (35) | 56 (38) | |
| If performed at home, patients may not present for routine primary care or follow-up for abnormal test results | 13 (9) | 39 (27) | 49 (33) | 46 (31) | |

that rescheduling well care was often unsuccessful: "I'll have the medical assistant call… but we have a really high no-show rate when people are just coming in for well exams" (APP, Women's Health).

Clinicians also noted that patients were afraid to come for care early in the pandemic: "Patients were hesitant, especially in the first year of [the] COVID pandemic, to leave their home for unnecessary reasons, including screening tests such as Pap smear" (MD, Family Medicine). Later in the pandemic, when more patients were seen for primary care, clinicians described situations where other medical conditions took priority: "primary care visits were all like trying to catch up on everything else cause all of a sudden now everyone's diabetes is out of control, and their anxiety is out of control, and cancer screening ends up being at the bottom of the list among the issues that they want to talk about" (MD, Family Medicine). As the pandemic moved into the endemic phase, clinicians described additional challenges: "The social determinants are still hitting some of our patients pretty hard… I don't know that it's COVID as much anymore that's affecting their ability to access care" (MD, Family Medicine).

**Table 7.** Qualitative themes with exemplar quotes.

| Theme | Exemplar quotes |
| --- | --- |
| Initial pandemic-associated barriers | "I would say it definitely disrupted all the cancer screenings, the mammo[gram]'s, the colonoscopies, the pap smears, I would say for the whole year of 2020 into about March of 2021." (APP, Family Medicine)<br>"We were only doing acute visits… everything else was by phone." (MD, Family Medicine) |
| Ongoing barriers (system and staffing) | System-related:<br>"We have the EMR triggering, and we have active tracking of abnormal Paps. But as far as getting people in for their routine screening, I don't believe we have someone actively tracking that. I feel like it's more on the provider picking it up as they open the chart." (APP, Family Medicine)<br>Staffing-related:<br>"We are still working with reduced staff in the office. So, there are definitely still much fewer appointments available." (APP, Family Medicine)<br>"We realized … we really need to start doing colposcopy again. But unfortunately, that's also when our physician colposcopy provider left." (MD/DO, Family Medicine)<br>"Rates of burnout, and then the competition from other systems, hiring people away was pretty debilitating at times." (APP, Family Medicine) |
| Facilitators and strategies for catching up on cervical cancer screening | Staffing and tracking:<br>"Patients get reminders… the health center as a whole has been trying to run lists of people that are due and bring them in." (APP, Family Medicine)<br>"If they had an abnormal PAP, the nursing staff would have ticklers [in the EMR] created as a reminder that it's time for the patient to have a PAP… We have two nurses who are dedicated not for just PAP tracking but for general ticklers." (MD/DO, Internal Medicine).<br>HPV self-sampling benefits:<br>"It decreases any concerns for like privacy, for discomfort, you know, patients who have trauma histories, maybe patients who are transgender, patients who, you know, like I said, work schedules don't allow them to get in on time, um, it just opens up a way for them to still all be screened in a way that can hopefully feel comfortable and accessible." (APP, OBGYN/Women's Health)<br>"I think it could be [useful to address pandemic-related screening deficits]. Especially if we don't have, um, as many in-person appointments available." (APP, Family Medicine)<br>HPV self-sampling concerns:<br>Inadequate sample:<br>"Making sure that people you know, kind of collect it correctly, mostly just because in my experience, people have not great knowledge about their own anatomy sometimes… if somebody accidentally puts the swab in their rectum, instead of the vagina, you would probably get an HPV result, because you can do HPV testing in the rectum, but you're not getting a, a cervical cancer screening." (APP, OBGYN/Women's Health)<br>Kits will not be returned:<br>"We do our –occult blood sampling with home tests, and sometimes –many times, those kits go home and never come back. We're always chasing a patient to kind of get them to bring it back or mail it back." (APP, Family Medicine) |

## Ongoing barriers (system and staffing)

Several participants described current and ongoing limitations to existing systems: "Only if a patient has had an abnormal [result] are they actively being tracked… [otherwise] until they access the Health Center for their next visit we really have no idea" (APP, Women's Health/OBGYN). Others described EMR functionality that went unused due to limited staff capacity or poorly functioning EMRs: "In our old system you could literally put a quick text [smart phrase that pulls patient information into a medical record note]… and it will just come up with all the history of the Paps. We can't do any of it in this new system… I'm literally going through the system, and looking at all the past Paps, and I'm writing them in the note" (MD, Family Medicine).

Participants described profound staffing shortages: "We're missing MAs, front desk, providers, nurses too. Pretty much literally everybody, every position, we're short" (APP, Family Medicine). Another said: "We stayed [open] without somebody cleaning the clinic 100%… so we had to do some of the work ourselves" (MD, Family Medicine). Staffing shortages also negatively impacted outreach: "We're not outreaching to patients and trying to get them in, we're just trying to get through the day… we just don't have the manpower to see everybody" (APP, Family Medicine). The relatively

lower compensation at FQHCs posed an additional challenge both to staff retention and to creating and utilizing patient tracking systems: "As a federally qualified health center, we often are not the best payer for different roles. And so we tend to have a lot of turnover, particularly in our medical assistants, nurses, and it's quite hard to hire." Additionally, this participant also noted, "We also tend not to have the biggest or the most robust IT department… And any time we need to get information from these registries, we need to ask our IT department. But they're pretty understaffed. And also underpaid" (MD, Family Medicine). Childcare also posed challenges: "I'd say the majority of our staff in the nursing and medical assistant roles are moms and some of them are single moms. So we lost a few because… they had no childcare [realted to the pandemic] or they couldn't come in" (APP, Family Medicine). In contrast, COVID-19 vaccine mandates were not felt to be significant contributors to staff shortages.

## Facilitators and strategies for catching up on cervical cancer screening

The participants discussed how the availability of COVID-19 vaccinations shifted the risk-benefit ratio of seeing patients in person for routine care: "before we were able to be vaccinated… it felt like unnecessary risk" (APP, Women's Health). As the pandemic continued into its second year, clinicians perceived the benefits of resuming in person visits outweighed the risk of contracting COVID-19 in healthcare settings; therefore, the focus shifted to catch-up measures: "When we realized that this was gonna be a long-term change… there was a big push to catch people up [with screening for cervical cancer]" (APP, Family Medicine).

Participants discussed strategies for patient outreach to catch-up on screening, including automated components within the EMR, dedicated staff who identify patients who are due to screen, providing evening or weekend hours, and mobile health units. One noted, "The health center as a whole has been trying to run lists of people that are due and bring them in" (APP, Family Medicine). Clinicians described strategies related to accountable care organizations, which are value-based care entities promoted by the US Centers for Medicare and Medicaid (*Centers for Medicare & Medicaid Services, 2023*), stating: "We're an accountable care organization, it incentivizes getting all of your quality metrics where you want them… The pap smears are tracked every quarter… If you hit above 75% of your pap smears, they give you an incentive quarterly" (APP, Family Medicine). Another suggested that healthcare systems and insurance plans could be utilized: "We [our practice] discussed perhaps using our accountable entity to try to do some outreach as well, because they do outreach right now for colon cancer and mammograms" (MD, Family Medicine).

Some participants described potential strategies to increase staff retention: "Increase in pay I feel will help. But also recognition for the staff, because some of the staff feel underappreciated…. and maybe more organized so that everything can run smoothly and uniformly" (APP, Family Medicine). Another added: "Better salaries, better benefits, better working conditions. In the sense that if somebody needed to take care of a child and go home early, then staggered staffing, flexible hours as part of the benefits, so that somebody else can cover. And, of course, monetary, icing on the cake, so to speak, always works" (MD, OBGYN).

Self-sampling for HPV testing is not currently FDA approved in the US, but may be an option in the future. Most participants thought self-sampling would be helpful to address pandemic-related screening deficits: "People are coming back with a lot of problems that they've been hanging on to for a couple of years. So that could help take care of some of their health maintenance and not further delay it because they're worried about X, Y, Z also. Then sure, that would help with the COVID deficit specifically" (APP, Family Medicine). Many noted that patients self-collected other specimens, and felt that HPV self-collection would be feasible: "We have a lot of our patients doing self-swabs right now anyways for vaginitis… and I'm used to having patients swab themselves for other things like in pregnancy we do GBS swabs, so I feel confident that people can correctly be instructed on how to self-swab" (APP, OBGYN). However, others were concerned about patients' abilities to properly collect the specimens: "There's certain populations, especially the underserved community that I do work in might face challenges to follow the instruction or even read on how to do it" (MD/DO, Family Medicine). Others described negative experiences using mailing for self-collected colon cancer screening: "It would be really clever if we could just send out swabs to patients. But I don't know. We tried that with FIT (fecal occult blood) testing, and we were told by the lab that they don't get a high enough return of the kits. And so it actually was cost prohibitive to just be sending out FIT tests" (MD/DO, Family Medicine).

## Discussion

We examined patterns of cervical cancer screening provision and abnormal results follow-up between October 2021 through July 2022 among clinicians practicing in federally qualified health centers. Over 80% of clinicians reported decreased screening during the start of the pandemic in 2020, but approximately 67% reported that screening had resumed to pre-pandemic levels at the time of the survey (2021–2022). Those who identified specialty as family medicine or other had decreased odds for, and those who identified training as APP, had increase odds for performing the same or more screening at time of survey (2021–22) as compared to before the pandemic. Clinician barriers, both reported quantitatively and qualitatively, focused on staffing shortages as well as structural systems to track and reestablish care for those who were overdue for screening and those who needed follow-up after an abnormal screening test.

Barriers to screening evolved over the course of the pandemic. In 2020, fear of contracting COVID was the primary barrier to provision of services by clinicians and health systems, and use of services by patients. Clinicians described near cessation of cervical cancer screening services early in the pandemic, as both clinicians and patients felt that the risk of contracting COVID when providing well care outweighed the benefits of cervical cancer screening in the short term. Vaccinations and the realization that COVID was becoming endemic changed this calculus, and clinics began re-opening services and recalling patients for screenings. In 2021/22, the primary barrier to cervical cancer screening shifted from contagion concerns to staffing shortages and the need for primary care clinicians to address other chronic health conditions. However, clinicians also noted that patients not scheduling or not attending appointments was an important barrier to screening. Quantitative findings indicated that cancer screenings were less often performed in specialties that did not focus on women's health, such as internal or family medicine. Qualitative data indicated that this may have resulted from a need to provide direct care for COVID-19 patients or to focus on other chronic health conditions that had worsened due to lack of care during 2020 (*Amit et al., 2020*; *Castanon et al., 2021*; *Network EHR, 2020*). In addition, our findings noted that APPs performed more cervical cancer screening than physicians, which could indicate appropriate allocation of patients needing preventive care to APPs, while assigning sicker patients to physicians who could better address complex medical concerns. Additional research is needed to confirm and further explore these findings.

Staff shortages hindering the ability to provide cervical cancer screening and follow-up care were reported by nearly half of clinicians. Clinicians reported reductions in staffing at all levels: physicians/APPs, nurses, medical assistants, and front desk staff. Staff shortages, both clinical and non-clinical across many healthcare settings, have been reported in other contexts as a result of the pandemic (*Holthof and Luedi, 2021*; *Chervoni-Knapp, 2022*). Two factors were felt to be the most important contributors to staff shortages: low salaries and lack of childcare. Because FHQCs typically pay lower salaries than other practice settings (*Friedberg et al., 2017*; *Quinn et al., 2013*), participants reported high levels of staff turnover and difficulties with recruitment. Pandemic-related remote schooling and rules related to infection control created childcare difficulties for many parents. Participants reported this to be a particular problem for female staff in lower salaried positions, such as medical assistants (*Boesch and Hamm, 2020*; *Organisation for Economic Co-operation and Development, 2019*).

Strategies for addressing pandemic-related screening deficiencies included improving staffing levels as well as systems for follow-up and tracking. Several clinicians described success associated with robust tracking systems including population management reports, system-wide incentives, automated patient outreach, and dedicated staff for patient recall and scheduling. Others, however, reported absent systems or being unable to utilize EMR capabilities due to staff shortages. Higher salaries, improved organization within the healthcare system, and ensuring that staff felt respected and valued by leadership were felt to be important strategies for improving care provision (*Prasad et al., 2021*; *Serrano et al., 2021*; *Sinsky et al., 2021*; *Talbot and Dean, 2018*).

Participants overall felt that HPV self-sampling would be a useful tool to address pandemic-related screening deficits, as has been noted in the literature (*Fuzzell et al., 2021*). Many felt confident that patients could self-collect the swabs given their experience using self-swabbing with patients for vaginitis or group B strep in pregnancy. However, others were concerned that patients might not collect the specimen properly, leading to a false negative cancer screening result. Self-sampling when using PCR-based testing has demonstrated overall similar accuracy to clinician-based samples (*Arbyn et al., 2022*), though studies to validate this in US populations are ongoing (*National Cancer Institute,*

*2023*). For some participants, clinic-collected sampling was viewed more favorably than home-testing via mailed kits due to negative experiences with home-based colon cancer testing. A meta-analysis of self-sampling indicated increased screening participation when self-sampling is offered, with clinic-based offering being more effective than mail-in kits (*Costa et al., 2023*).

As healthcare continues to face challenges including COVID-19, influenza, behavioral health, and exacerbation of chronic diseases, strategies are needed to ensure that patients are provided with cervical cancer prevention services. This is especially important in FQHCs, who serve patients at the highest risk of invasive cervical cancer (*Hébert et al., 2018*; *Singh et al., 2004*; *Barry and Breen, 2005*; *Friedman et al., 2012*; *Bradley et al., 2001*). Maintaining adequate staffing is a critical need noted in our study and by others (*Frogner, 2022*). Higher salaries were felt to be most important, as well as improved organization of clinic function and flexible scheduling to support working parents with childcare needs (*Burrowes et al., 2023*; *U.S. Bureau of Cancer Statistics, 2023*).

This study has several strengths and weaknesses. We surveyed clinicians practicing in FQHCs in the US on the perceived impact of the pandemic on screening and abnormal results follow-up. Few investigations thus far have examined perceptions of those practicing in FQHCS, in particular as it pertains to impacts of pandemic-related challenges to cervical cancer screening. Despite this, we note several limitations. We recruited our sample through FQHC networks; thus, we were unable to calculate a response rate, nor were we able to achieve a nationally representative sample and thus, findings cannot be widely generalized. Notwithstanding efforts to achieve a regionally diverse sample, 63% of responding clinicians were practicing in the Northeast at the time of their participation. Given that COVID-19 policies varied widely by state, this regional imbalance may limit the generalizability of our results. Despite the oversample of clinicians in the Northeast, region was not a significant predictor of either outcome. Similarly, our sample was 85% female and 70% White. Although ideally we would have included a sample that was more diverse with respect to race and gender, these characteristics are not disparate from the majority of clinicians who perform cervical cancer screening (e.g., race: Women's Health NPs [77% White] (*Healthcare Ws, 2018*), active Ob/Gyns [67% White] (*AAMC, 2022*), all active physicians [64% White] (*AAMC, 2022*); gender: all NPs [92% female] (*Hooker et al., 2016*), Ob/Gyns [64% female] (*AAMC, 2022*), all active physicians [37% female] (*AAMC, 2022*)). Importantly, we do not have data on the overall number of screenings provided by each FQHC. The majority of our sample reported that they personally were providing screening at pre-pandemic levels, but half also report staff shortages impacting screening and follow up. Therefore, we cannot confirm whether the efforts of remaining staff are sufficient to compensate for missing personnel in terms of the overall availability of services. Finally, the use of manual forward selection with our a priori determined significance level has limitations, including the possibility of overfitting. Additional studies would be useful to confirm these findings.

These findings highlight that in late 2021 and early 2022, clinicians in FQHCs are still perceiving impacts of the pandemic broadly to cervical cancer screening. They also still report experiencing pandemic-related impacts of staffing changes on screening and follow-up. If not addressed, reductions in screening due to staff shortages, and low patient engagement with the healthcare system may lead to increase in cervical cancer in the short and long term. Future research should closely track trends in provision of screening, colposcopy, and treatment services in underserved communities and settings in order to avoid future increases in cancer incidence.

## Acknowledgements

The authors acknowledge Moffitt Cancer Center's Biostatistics and Bioinformatics Shared Resource (BBSR).

## Additional information

### Funding

| Funder | Grant reference number | Author |
| --- | --- | --- |
| American Cancer Society | | Susan T Vadaparampil |

| Funder | Grant reference number | Author |
|--------|------------------------|--------|

The funders had no role in study design, data collection and interpretation, or the decision to submit the work for publication.

## Author contributions

Lindsay Fuzzell, Conceptualization, Data curation, Supervision, Visualization, Writing – original draft, Project administration, Writing – review and editing; Paige Lake, Formal analysis, Visualization, Writing – original draft, Project administration, Writing – review and editing; Naomi C Brownstein, Conceptualization, Data curation, Formal analysis, Supervision, Visualization, Methodology; Holly B Fontenot, Conceptualization, Formal analysis, Funding acquisition, Investigation, Methodology, Writing – review and editing; Ashley Whitmer, McKenzie McIntyre, Project administration, Writing – review and editing; Alexandra Michel, Sarah L Rossi, Formal analysis, Visualization, Writing – review and editing; Sidika Kajtezovic, Formal analysis, Visualization; Susan T Vadaparampil, Conceptualization, Resources, Data curation, Supervision, Funding acquisition, Investigation, Methodology, Writing – review and editing; Rebecca Perkins, Conceptualization, Resources, Data curation, Formal analysis, Supervision, Funding acquisition, Investigation, Visualization, Methodology, Writing – original draft, Writing – review and editing

## Author ORCIDs

Lindsay Fuzzell ⓘ https://orcid.org/0000-0001-9688-5365
Paige Lake ⓘ https://orcid.org/0000-0002-5591-6417

## Ethics

This study was approved by Moffitt Cancer Center's Scientific Review Committee and Institutional Review Board (MCC #20048) and Boston University Medical Center's Institutional Review Board (H-41533).

## Decision letter and Author response

Decision letter https://doi.org/10.7554/eLife.86358.sa1
Author response https://doi.org/10.7554/eLife.86358.sa2

# Additional files

## Supplementary files
- MDAR checklist
- Reporting standard 1.

## Data availability

Full human subjects data are unavailable via a data repository due to confidentiality concerns. A limited dataset may be made available upon reasonable request from other academic researchers and requests should be submitted via email to the corresponding author and will be approved on a case by case basis by study PIs and the institutional SRC and IRB. SAS version 9.4 was used to analyze data. SAS code has been made available at https://doi.org/10.7910/DVN/URBYSD.

The following dataset was generated:

| Author(s) | Year | Dataset title | Dataset URL | Database and Identifier |
|-----------|------|---------------|-------------|--------------------------|
| Fuzzell L | 2023 | CC PROGRESS TLC SAS code | https://doi.org/10.7910/DVN/URBYSD | Harvard Dataverse, 10.7910/DVN/URBYSD |

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
