## [Editor Report]

This US study presents findings from an online survey and in-person interviews of healthcare providers in areas associated with cervical screening provision during the post-acute phase of the pandemic. The findings are valuable as they provide insights into a range of areas, from healthcare characteristics to screening barriers and HPV self-sampling. The evidence supporting the claims of the authors is solid. The work will be of interest to public health scientists and a cancer prevention and control audience.

---

## [Decision Letter]

**Decision letter after peer review:**

Thank you for submitting your article "Examining the impact of the COVID-19 pandemic on cervical cancer screening practices among clinicians practicing in Federally Qualified Health Centers: A mixed methods study" for consideration by *eLife*. Your article has been reviewed by 3 peer reviewers, and I oversaw the evaluation in my dual role of Reviewing Editor and Senior Editor. The following individual involved in the review of your submission has agreed to reveal their identity: Parker Tope (Reviewer #1).

Essential revisions:

As is customary in *eLife*, the reviewers have discussed their critiques with one another. What follows below is an edited compilation of the essential and ancillary points provided by reviewers in their critiques and in their interaction post-review. Please submit a revised version that addresses these concerns directly. Although we expect that you will address these comments in your response letter, we also need to see the corresponding revision clearly marked in the text of the manuscript. Some of the reviewers' comments may seem to be simple queries or challenges that do not prompt revisions to the text. Please keep in mind, however, that readers may have the same perspective as the reviewers. Therefore, it is essential that you attempt to amend or expand the text to clarify the narrative accordingly.

*Reviewer #1 (Recommendations for the authors):*

Introduction

The introduction contextualizes well previous literature on the impact of the pandemic on cancer screening services and introduces tangible examples of how the pandemic influenced factors along the screening continuum.

While the authors mention investigating the association between clinician characteristics and outcomes of interest in the methods, there is little justification for doing so. Are clinical characteristics meant to serve as proxies for determining the patient populations that are attended to by clinician participants?

Methods

I would suggest reorganizing the methods into the following sections for logical flow: Target population, Survey development and validation (which would include measures), survey administration (which would include Participant Recruitment), and finally analysis (Quantitative and Qualitative). This would aid the reader in following the process of the survey construction, dissemination, and finally, how obtained data were used.

What is the target population of the study? Explicitly stating the target population can help readers determine if sampling methods were appropriate given the sampling frame.

The authors mention federally qualified health centres (FQHC) as their participant sources, however, there is no explanation as to why participant recruitment was focused on FQHCs. Are there key populations served by FQHCs in the US? Including further explanation would allow for further assessment of the study's generalizability as well as comparability with other research outside of the US.

How was the survey administered? Was there a particular software or an electronic form used for the online survey? I suggest including this information in the Methods section for transparent reporting.

The authors state that the survey questions were piloted across an expert panel, however, it is unclear as to whether the platform that was used for the survey administration was also piloted for possible technical functionality issues. Not piloting the technology could lead to unknown sources of error and thus introduce bias. If piloting was conducted, I suggest that this be more clearly stated in the methods.

The authors collect race/ethnicity information from their survey participants, specifically collecting disaggregated race/ethnicity data. In their regression analyses, the researchers aggregated racial categories without explanation. In accordance with best practices in handling race/ethnicity data, when collecting disaggregated data, subsequent aggregation of race/ethnicity categories should be clearly justified (e.g., insufficient sample size).

There is no mention of missing data, or incomplete survey responses. Did the authors collect this information (i.e., whether survey responses that were started but not completed were captured in their final dataset?). Was data cleaned and assessed for unlikely or aberrant values?

With regards to the qualitative interviews, how were participants interviewed? In-person, via Zoom, or over the phone? Who conducted the interviews?

I have difficulty understanding why the authors conducted multivariable logistic regression, rather than univariate. Given the study design (i.e., survey) as well as the main objective of the study, which was to explore the perceived need for screening and appears to be descriptive, I am uncertain about the authors' adjustment for several covariates. The overarching concern is that adjustment for covariates (unclear as to whether these are theorized as predictors or confounders) conflates a descriptive research question with causal methods. If the objective were to determine how clinician characteristics causally affected (1) the level of screening pre- and during-pandemic times and (2) the severity of systemic difficulties (i.e., staffing), then adjustment for covariates should be justified and reflected in the study's objective. There appears to be little causality to be inferred through this study and rather a descriptive perspective, which would only appropriately justify univariate, descriptive analyses.

Even if adjustment for covariates were appropriate for this research question, the use of manual forwards selection using p-value as the selection criteria can lead to over-fitting and requires well-justified selection of potential covariates to be sequentially added to the model. If the authors proceed to include previously conducted analyses in this manuscript, I suggest that such limitations be acknowledged.

Results and Discussion:

The authors comprehensively present the descriptive findings of the survey data. I very much appreciated the inclusion of direct quotes from interviewees in addition to summarizing key qualitative pieces in the included table. These quotes provide a narrative component to the article and give voice to the challenges and frustrations experienced by clinicians.

Given the authors' emphasis on interviewees' difficulties in staffing, accommodating childcare for staff, and better compensating non-provider healthcare workers, I would suggest the inclusion of a section in the discussion emphasizing how the COVID-19 pandemic has also disproportionately affected the lived experiences of non-providers in healthcare, who have essential roles in facilitating the cancer care system.

*Reviewer #2 (Recommendations for the authors):*

Introduction:

1. Provide a brief overview of what Federally Qualified Health Centers (FQHCs) are and how they differ from other healthcare facilities in the US. It would help readers unfamiliar with the US healthcare system understand why safety net facilities like FQHCs are essential in cervical cancer screening.

2. Explain why the chosen period (October 2021 through July 2022) was significant or relevant to the study. This would help readers understand why this specific time frame was chosen for the study and how it might have impacted the findings. Was this period extra hard in the U.S?

Method:

3. The paper would benefit from discussing the choice of statistical method for analysing the quantitative survey data. Stepwise regression is a discussed method with the downside of overfitting. At the same time, an explanation or discussion should follow of choosing a significance value for entry at 0.10.

4. Why did the authors choose the p-values of 0.10 as significant?

Results:

5. Page 7, line 190: The author points out that “the most commonly reported barriers were limited in-person appointment availability (46%)…” However, this number cannot be found in Table 3, pages 20-21. I guess that it is a typo and should instead be 45%?

Discussion:

6. Discuss whether the composition of responders represents the people who generally work at the safety net facilities. The sample contains an overrepresentation of white females, which could affect the results.

7. The paper would benefit from a discussion on the choice of statistical method for analysing the quantitative survey data. It is my understanding that stepwise regression is a discussed method with the downside of overfitting. At the same time, there should follow an explanation or discussion of the choosing of a p-value of 0.10, as this in my opinion is high.

Tables and figures:

8. In Table 1, under age, the sum of the numbers does not add up to N 148; instead, the sum is 147.

*Reviewer #3 (Recommendations for the authors):*

1. General: inconsistencies in percentages between the manuscript text and tables were observed throughout. The manuscript needs to be checked carefully and corrections made. Some may be due to a lack of rounding; appropriate rounding should be applied on percentages noted in tables and footnoted.

2. Abbreviations are provided in the text (and abstract) without defining these in the first place. These may be familiar/standard in the US but not for an international audience.

3. Title: only 45% of the participants of this study were clinicians. Adding or replacing this term with 'health care providers' would more accurately describe study participants. This point should be applied throughout the article.

4a. General/abstract: although I appreciate the constraints of the word limit for the abstract, the current wording does not do justice to the work presented. Suggest re-writing sections of it.

4b. Abstract/methods section: lines 35-38 are not methods but results. Other information should be stated in this section e.g. how the national sample was obtained, how the survey was conducted, and domains of questioning.

4c. Abstract/results: Findings in the Results section for APPs and ethnicity did not reach statistical significance as presented in the paper. There were various interesting findings that could replace these statements.

4d. Abstract/conclusion: although I agree with the validity of the statement in the conclusion, it does not sum up the results presented.

5a. Results section

Line 170: it is stated that 38% reported suspension of colposcopy and 6% of LEEP services, based on denominators of 95 and 127 participants respectively, after taking the number of unsure answers out of the total of 148 participants (as per footnote of Table 1). However, Table 1 also stated that only 115/148 provided colposcopy on site and 46/148 provided LEEP which has not been taken into consideration. Please revise both the manuscript text and Table 1 entries accordingly.

5b. Line 180: the p-value has been incorrectly rounded.

5c. Lines 185-188: The text states that clinician training was significantly associated with increased odds of the same or more screening however the p-value provided is 0.06 which signifies weak evidence at best. Even 0.05 is considered borderline significance. The same applies to the association with clinician race/ethnicity. Please amend these statements accordingly or remove.

6a. Discussion general: findings of non-attendance and increased frequency of women not booking screening appointments (not even mentioned in the results but presented in Table 3), are important points to mention in the discussion and linked to observed cervical screening attendance in the US reported during a similar time period.

6b. Lines 327-328: Is this statement based on qualitative evidence? If so please include this in the Results section as well.

6c. Lines 331-333: Quantitative findings referred to in this sentence were not included in the Results section nor relevant tables. It would be informative to provide a breakdown of screens provided by speciality in Table 1.

6d. Lines 335-336: The statement that APPs performed more screens than physicians has not been included in the results. It would be informative to provide a breakdown of screens provided per training in Table 1.

6e. Lines 367-369 These themes have already been raised earlier in the discussion (lines 339-346). Suggest merging the two relevant paragraphs.

7. Table 1: No details on staffing are provided in this table; title 9 rows from the end of the table should be amended.

8a. Table 2: recommend adding zeros before the point for more clarity.

8b. Table 2: a footnote listing the variables for which regression was adjusted should be listed.

8c. Table 3: add 'adjusted' to 'odds ratio'.

---

## [Author Response]

Essential revisions:Reviewer #1 (Recommendations for the authors):IntroductionThe introduction contextualizes well previous literature on the impact of the pandemic on cancer screening services and introduces tangible examples of how the pandemic influenced factors along the screening continuum.While the authors mention investigating the association between clinician characteristics and outcomes of interest in the methods, there is little justification for doing so. Are clinical characteristics meant to serve as proxies for determining the patient populations that are attended to by clinician participants?

Thank you for your comment. We intentionally sought to examine clinician characteristics that may be associated with perceived changes in cervical cancer screening and the impact of pandemic-related staffing changes on screening and abnormal results follow-up during the post-acute pandemic period. The reason for doing so was to identify characteristics that could be targets for future interventions or additional support. For example, if more family medicine practitioners reported lower screening rates, that indicates a potential need for interventions focused on family medicine clinicians to help to avoid future disparities in cervical cancer. Similarly, characteristics like age, race, ethnicity, gender, region are worth exploring as statistically significant associations and could indicate that more supports and resources could be provided for providers in particular sub-groups. We now include a sentence on pg. 4 that indicates the reasoning behind exploring these associations:

“In order to identify characteristics that could be targets for future interventions or additional supports, this paper examines the association of clinician characteristics with perceived changes in cervical cancer screening and the impact of pandemic-related staffing changes on screening and abnormal results follow-up during the pandemic period of October 2021 through July 2022 in FQHCs and safety net settings of care.”

MethodsI would suggest reorganizing the methods into the following sections for logical flow: Target population, Survey development and validation (which would include measures), survey administration (which would include Participant Recruitment), and finally analysis (Quantitative and Qualitative). This would aid the reader in following the process of the survey construction, dissemination, and finally, how obtained data were used.

We have addressed these changes as suggested on pgs. 4 and 5.

What is the target population of the study? Explicitly stating the target population can help readers determine if sampling methods were appropriate given the sampling frame.

The target population was clinicians who conducted cervical cancer screening in federally qualified health centers and safety net facilities in the United States during the post-acute phase of the COVID-19 pandemic, which is now noted on pg. 4 of the manuscript:

“The target population were clinicians, defined for the purpose of this study as physicians and Advanced Practice Providers (APPs), who conducted cervical cancer screening in federally qualified health centers and safety net settings of care in the United States during the post-acute phase of the COVID-19 pandemic.”

The authors mention federally qualified health centres (FQHC) as their participant sources, however, there is no explanation as to why participant recruitment was focused on FQHCs. Are there key populations served by FQHCs in the US? Including further explanation would allow for further assessment of the study's generalizability as well as comparability with other research outside of the US.

The study focused on safety net settings of care. The most common safety net setting in the US are FQHCs, federally funded health centers or clinics that serve medically underserved areas and populations and often provide care at no or low cost to those with limited or no health insurance. During the pandemic, there was little research that focused specifically on FQHCs and ability to provide cervical cancer screening to these underserved populations who are at higher risk of cervical cancer than the general population. This is noted on pg. 3 of the manuscript:

“Federally qualified health centers (FQHCs) in the US are government funded health centers or clinics that provide care to medically underserved populations. Maintaining cancer screening in these and other safety net facilities is critical as they serve patients at the highest risk for cervical cancer: publicly insured/uninsured, immigrant, and historically marginalized populations.”

How was the survey administered? Was there a particular software or an electronic form used for the online survey? I suggest including this information in the Methods section for transparent reporting.

The survey was administered via online survey designed and hosted via Qualtrics, a common platform for market and research surveys. This is now noted on pg. 4 of the manuscript:

“We recruited clinicians for participation in the online survey hosted via Qualtrics….”

The authors state that the survey questions were piloted across an expert panel, however, it is unclear as to whether the platform that was used for the survey administration was also piloted for possible technical functionality issues. Not piloting the technology could lead to unknown sources of error and thus introduce bias. If piloting was conducted, I suggest that this be more clearly stated in the methods.

Thank you for the opportunity to clarify the process. The survey items were piloted with the expert panel prior to design of the Qualtrics survey. Once survey items were embedded into the Qualtrics platform, the research team internally tested the survey, making note of any technical errors that resulted from skip logic, select all versus single selection items, etc., and correcting any issues that were identified. They study also used the same Qualtrics survey platform to design a separate survey for over 1200 providers in the year prior and thus, had extensive experience identifying technical issues. The testing of technical functionality by the study team is now noted on pg. 4 in the Survey development and validation section:

“The draft survey was reviewed by an expert panel of FQHC providers (n=8), refined, piloted, and finalized after incorporating pilot feedback and testing technical functionality of the Qualtrics survey among the study team.”

The authors collect race/ethnicity information from their survey participants, specifically collecting disaggregated race/ethnicity data. In their regression analyses, the researchers aggregated racial categories without explanation. In accordance with best practices in handling race/ethnicity data, when collecting disaggregated data, subsequent aggregation of race/ethnicity categories should be clearly justified (e.g., insufficient sample size).

The reviewer makes an excellent point. Due to small cell sizes for persons of color (n of less than 15 each for Black, Asian, mixed race, and other race categories), we elected to categorize this variable with two groups: white non-Hispanic versus all non-white races (including Hispanic/Latinx). This is noted on pg. 5 and we’ve added the reasoning to the text (small cell sizes):

“Race/ethnicity was categorized for analysis as white non-Hispanic versus all others due to small cell sizes of non-white and Hispanic participants.”

There is no mention of missing data, or incomplete survey responses. Did the authors collect this information (i.e., whether survey responses that were started but not completed were captured in their final dataset?). Was data cleaned and assessed for unlikely or aberrant values?

Data were cleaned and examined for potential duplicate responses identified by repeat IP address and identical participant characteristics, nonsensical free responses, or randomly selected responses. See added explanation now included in text on pg. 7:

“Data were cleaned and invalid surveys were removed. Invalid surveys included potential duplicate responses identified by repeat IP address, nonsensical write-in free responses, and those with numerous skipped items.”

Overall, in the Qualtrics survey platform, if participants skipped an item purposely or unintentionally left an item blank, Qualtrics automatically prompted them to complete that item. They could either choose to complete it or select “ignore” to skip the item. There were very few skipped or missing items, but if a particular participant appeared to be missing more than a few errant responses, all of their responses were assessed to determine whether their participation was invalid and whether their survey should be removed. For each item included in analyses, Tables 1, 3, 4, and 5 include a “valid N” column. The total N for this sample was 148, thus any valid Ns less than 148 indicate missing responses for that particular item.

With regards to the qualitative interviews, how were participants interviewed? In-person, via Zoom, or over the phone? Who conducted the interviews?

Interviews were conducted via Zoom by three co-authors trained in qualitative methodology. This info is now indicated on pg. 6:

“Interviews were conducted by three co-authors (RBP, AM, HBF) trained in qualitative methodology via video conference (Zoom)”

I have difficulty understanding why the authors conducted multivariable logistic regression, rather than univariate. Given the study design (i.e., survey) as well as the main objective of the study, which was to explore the perceived need for screening and appears to be descriptive, I am uncertain about the authors' adjustment for several covariates. The overarching concern is that adjustment for covariates (unclear as to whether these are theorized as predictors or confounders) conflates a descriptive research question with causal methods. If the objective were to determine how clinician characteristics causally affected (1) the level of screening pre- and during-pandemic times and (2) the severity of systemic difficulties (i.e., staffing), then adjustment for covariates should be justified and reflected in the study's objective. There appears to be little causality to be inferred through this study and rather a descriptive perspective, which would only appropriately justify univariate, descriptive analyses.

The reviewer is correct that the analyses presented are not causal and the aim was to examine cross-sectional associations between clinician characteristics (race/ethnicity, gender, age, region, clinician training, clinician specialty) and the dependent variable (perceived screening practices at the time of survey participation the same/more versus less than pre-pandemic). Our team, including a biostatistician (NB), conducted these analyses with these goals in mind. We also confirmed there were no mentions of causality in the manuscript. Multivariable logistic regression is appropriate because we aimed to examine associations of multiple independent variables with a single dependent variable (perceived screening practices), and each association controls for confounding by the other variables in the model. If needed, we can provide univariate crude associations by request for supplementary material.

Even if adjustment for covariates were appropriate for this research question, the use of manual forwards selection using p-value as the selection criteria can lead to over-fitting and requires well-justified selection of potential covariates to be sequentially added to the model. If the authors proceed to include previously conducted analyses in this manuscript, I suggest that such limitations be acknowledged.

We acknowledge that manual forward selection has drawbacks, as do other approaches such as backward selection or AIC. With a small sample size (N=148), we chose manual forward selection because it begins with a null model and builds upon itself incrementally, rather than backward selection, which starts with a larger model. The biostatistician and study team carefully selected variables to be added to the model. The limitations of manual forward selection are now noted on pg. 16 in the limitations section of the discussion:

“Finally, the use of manual forward selection with our a priori determined significance level has limitations, including the possibility of overfitting. Additional studies would be useful to confirm these findings.”

Results and Discussion:The authors comprehensively present the descriptive findings of the survey data. I very much appreciated the inclusion of direct quotes from interviewees in addition to summarizing key qualitative pieces in the included table. These quotes provide a narrative component to the article and give voice to the challenges and frustrations experienced by clinicians.Given the authors' emphasis on interviewees' difficulties in staffing, accommodating childcare for staff, and better compensating non-provider healthcare workers, I would suggest the inclusion of a section in the discussion emphasizing how the COVID-19 pandemic has also disproportionately affected the lived experiences of non-providers in healthcare, who have essential roles in facilitating the cancer care system.

Thank you for this thoughtful feedback. On pgs. 14-15, we include discussion of staff shortages (both clinical and non-clinical staff) as a result of the pandemic, and the impact on these workers.

“Staff shortages hindering the ability to provide cervical cancer screening and follow-up care were reported by nearly half of clinicians. Clinicians reported reductions in staffing at all levels: physicians/APPs, nurses, medical assistants, and front desk staff. Staff shortages, both clinical and non-clinical across many healthcare settings, have been reported in other contexts as a result of the pandemic.^29,30^ Two factors were felt to be the most important contributors to staff shortages: low salaries and lack of childcare. Because FHQCs typically pay lower salaries than other practice settings,^31,32^ participants reported high levels of staff turnover and difficulties with recruitment. Pandemic-related remote schooling and rules related to infection control created childcare difficulties for many parents. Participants reported this to be a particular problem for female staff in lower salaried positions, such as medical assistants.^33,34^”

Reviewer #2 (Recommendations for the authors):Introduction:1. Provide a brief overview of what Federally Qualified Health Centers (FQHCs) are and how they differ from other healthcare facilities in the US. It would help readers unfamiliar with the US healthcare system understand why safety net facilities like FQHCs are essential in cervical cancer screening.

Excellent suggestion. We now describe FQHCs in the second paragraph of the introduction, contextualized by the higher rates of those diagnosed with cervical cancer in the populations served by FQHCs.

2. Explain why the chosen period (October 2021 through July 2022) was significant or relevant to the study. This would help readers understand why this specific time frame was chosen for the study and how it might have impacted the findings. Was this period extra hard in the U.S?

We began recruitment in October 2021 because at that time, the pandemic appeared to be less acute and COVID-19 vaccination had become widespread in the US, with healthcare organizations attempting to resume normal operations. The goal was to recruit approximately 150 clinicians. However, the Omicron wave hit in winter 2021-22, which overwhelmed healthcare systems and forced us to pause recruitment until approximately March 2022. We resumed recruitment efforts in the spring of 2022 and reached our target sample size by July. We have added information to the Methods section (Participant recruitment and target population) on pg.4 indicating the goal to focus on perceived cervical cancer screening practices during the post-acute period after vaccination was generally available.

Method:3. The paper would benefit from discussing the choice of statistical method for analysing the quantitative survey data. Stepwise regression is a discussed method with the downside of overfitting. At the same time, an explanation or discussion should follow of choosing a significance value for entry at 0.10.

We now acknowledge the limitation of overfitting on pg. 16 of the manuscript: “the use of manual forward selection with our a priori determined significance level has limitations, including the possibility of overfitting” Additionally, because of the small sample size, the nature of these analyses is exploratory. Using a priori hypotheses would have included more potential variables and would have resulted in a larger model than what we ultimately utilized, with very small cell sizes for many of the variables. As suggested by the study biostatistician, we selected p of.10 as a significance value for entry. This strikes a balance between the commonly accepted method of using the AIC (Akaike’s Information Criterion), which implicitly assumes a significance level of 0.157, and simultaneously mitigates the potentially low power corresponding to a level of 0.05 in a sample as small as ours, which is now noted on pg. 6:

“We used manual forward selection with a value for entry and significance of 0.10 because this strikes a balance between the commonly accepted method of using AIC (which assumes significance level of 0.157), and the often used α of 0.05, which could lead to failure to identify associations due to small sample size.”

4. Why did the authors choose the p-values of 0.10 as significant?

Similar to our response above pertaining to entry p values of.10, we use a significance level of.10 to balance the tradeoff between high type I error inherent in other levels such as 0.157 (the level assumed when using the AIC to choose a model) and low power to conduct exploratory analyses with a smaller α. This small study was designed as a supplement to a larger quantitative study and was intended to be hypothesis generating for future, confirmatory studies. Given these goals, the study biostatistician and research team felt that 0.10 was an appropriate significance level selected a priori for our study. On pg. 6, we have clarified the reasoning stated in the response above applies to both entry and significance:

“We used manual forward selection with a value for entry and significance of 0.10 because…”

Results:5. Page 7, line 190: The author points out that “the most commonly reported barriers were limited in-person appointment availability (46%)…” However, this number cannot be found in Table 3, pages 20-21. I guess that it is a typo and should instead be 45%?

Thank you for noting this typo. We have corrected the percentage to 45% in text on pg. 9.

Discussion:6. Discuss whether the composition of responders represents the people who generally work at the safety net facilities. The sample contains an overrepresentation of white females, which could affect the results.

As we note in the public response, we acknowledge the high enrollment of White women in our provider sample and now address this point in the discussion on pg. 16:

“Similarly, our sample was 85% female and 70% White. Although ideally we would have included a sample that was more diverse with respect to race and gender, these characteristics are not disparate from the majority of clinicians who perform cervical cancer screening (e.g., race: Women’s Health NPs [77% White], active Ob/Gyns [67% White], all active physicians [64% White]; gender: all NPs [92% female], Ob/Gyns [64% female], all active physicians [37% female]).”

Data describing these characteristics are reported in the Association of American Medical Colleges (AAMC) 2022 Physician Specialty Data Report and Executive Summary, the 2018 NPWH Women’s Health Nurse Practitioner Workforce Demographics and Compensation Survey: Highlights Report, and a published paper describing the characteristics of nurse practitioners in the US, which are cited in text.

7. The paper would benefit from a discussion on the choice of statistical method for analysing the quantitative survey data. It is my understanding that stepwise regression is a discussed method with the downside of overfitting. At the same time, there should follow an explanation or discussion of the choosing of a p-value of 0.10, as this in my opinion is high.

As we state in our response to Reviewer 1, multivariable logistic regression is appropriate because we aimed to examine associations of multiple independent variables with a single dependent variable (perceived screening practices), and each association controls for confounding by the other variables in the model. On pg. 6 of the manuscript we state the reasoning for use of exact models:

“We conducted separate exact binary logistic regressions (due to small cell sizes)…”

As we noted in our response to Reviewer 2, we now acknowledge the limitation of overfitting on pg. 16:

“the use of manual forward selection with our a priori determined significance level has limitations, including the possibility of overfitting”. Additionally, as suggested by the study biostatistician, we selected p of.10 as a significance value for entry. This strikes a balance as it is more liberal than a commonly accepted method of using the AIC (Akaike’s Information Criterion), which implicitly assumes a significance level of 0.157, and simultaneously mitigates the potentially low power corresponding to a level of 0.05 in a sample as small as ours, which is now noted on pg. 6: We used manual forward selection with a value for entry and significance of 0.10 because this strikes a balance between the commonly accepted method of using AIC (which assumes significance level of 0.157), and the often used α of 0.05, which could lead to failure to identify associations due to small sample size.”

Tables and figures:8. In Table 1, under age, the sum of the numbers does not add up to N 148; instead, the sum is 147.

Thank you for your attention to detail. We confirmed with a check of our descriptive statistics results and one participant did not respond to this item. Therefore, the valid n for age should be 147 as now indicated in Table 1.

Reviewer #3 (Recommendations for the authors):1. General: inconsistencies in percentages between the manuscript text and tables were observed throughout. The manuscript needs to be checked carefully and corrections made. Some may be due to a lack of rounding; appropriate rounding should be applied on percentages noted in tables and footnoted.

Thank you for noting this. All errors have been corrected.

2. Abbreviations are provided in the text (and abstract) without defining these in the first place. These may be familiar/standard in the US but not for an international audience.

Thank you for noting this oversight. We have amended the text to spell out acronyms where appropriate.

3. Title: only 45% of the participants of this study were clinicians. Adding or replacing this term with 'health care providers' would more accurately describe study participants. This point should be applied throughout the article.

Thank you for your note on terminology. As defined in this study, all participants meet the definition of “clinician”. We have clarified this in the text on pg. 4:

“The target population were clinicians, defined for the purpose of this study as physicians and advanced practice providers, who conducted cervical cancer screening in federally qualified health centers in the United States during the post-acute phase of the COVID-19 pandemic.”

While we agree that the use of ‘health care provider’ is commonly used, we use ‘clinician’ throughout the manuscript because a pilot test of survey items for the parent study to this survey indicated that some physicians and patients may perceive the language of ‘provider’ to be paternalistic and potentially antisemitic. We therefore phrased survey items utilizing the term ‘clinician’ and this has carried over to our manuscripts. We are amenable to changing this language if the international readership of the journal would find this more appropriate.

4a. General/abstract: although I appreciate the constraints of the word limit for the abstract, the current wording does not do justice to the work presented. Suggest re-writing sections of it.4b. Abstract/methods section: lines 35-38 are not methods but results. Other information should be stated in this section e.g. how the national sample was obtained, how the survey was conducted, and domains of questioning.

We have restructured the method section of the abstract to reflect these suggestions.

4c. Abstract/results: Findings in the Results section for APPs and ethnicity did not reach statistical significance as presented in the paper. There were various interesting findings that could replace these statements.

We have amended this section of the abstract to better reflect our findings.

4d. Abstract/conclusion: although I agree with the validity of the statement in the conclusion, it does not sum up the results presented.

We have amended the concluding statement of the abstract as suggested.

5a. Results sectionLine 170: it is stated that 38% reported suspension of colposcopy and 6% of LEEP services, based on denominators of 95 and 127 participants respectively, after taking the number of unsure answers out of the total of 148 participants (as per footnote of Table 1). However, Table 1 also stated that only 115/148 provided colposcopy on site and 46/148 provided LEEP which has not been taken into consideration. Please revise both the manuscript text and Table 1 entries accordingly.

Thank you for noting this important oversight. We have recalculated the percentage that suspended colposcopy and LEEP based on new denominators of those who actually perform these services on site and have updated the percentages in the Results section of the manuscript and in Table 1.

5b. Line 180: the p-value has been incorrectly rounded.

We have corrected the p value as mentioned (now.04 rather than.03).

5c. Lines 185-188: The text states that clinician training was significantly associated with increased odds of the same or more screening however the p-value provided is 0.06 which signifies weak evidence at best. Even 0.05 is considered borderline significance. The same applies to the association with clinician race/ethnicity. Please amend these statements accordingly or remove.

Clinician training was associated with increased odds of the same or more screening, a finding that was statistically significant based on our significance level of 0.10, which was chosen a priori/before we examined the data. We acknowledge that this level could be considered borderline significant if one had chosen a different level, such as 0.05. Given the small sample size and exploratory nature of this study, we felt that this significance level was justified, and even more conservative than other options such as the AIC (α=0.157). We note in the discussion that these findings should be explored/confirmed with additional research and also note on pg. 6 of the method the reasoning for choosing 0.10 as the significance value as noted in the response to Reviewer 2. (A value of 0.10 strikes a balance as it is more liberal than a commonly accepted method of using the AIC (Akaike’s Information Criterion), which implicitly assumes a significance level of 0.157, and simultaneously mitigates the potentially low power corresponding to a level of 0.05 in a sample as small as ours.)

6a. Discussion general: findings of non-attendance and increased frequency of women not booking screening appointments (not even mentioned in the results but presented in Table 3), are important points to mention in the discussion and linked to observed cervical screening attendance in the US reported during a similar time period.

Thank you for pointing this out. We have added the following to the discussion on pg. 14:

“However, clinicians also noted that patients not scheduling or not attending appointments was an important barrier to screening.”

6b. Lines 327-328: Is this statement based on qualitative evidence? If so please include this in the Results section as well.

The refenced statement states: “Clinicians described near cessation of cervical cancer screening services early in the pandemic, as both clinicians and patients felt that the risk of contracting COVID when providing well care outweighed the benefits of cervical cancer screening in the short term,” is based on both quantitative and qualitative evidence. Most (80%) of survey respondents noted decreased screening early in the pandemic. This was contextualized in qualitative interviews as near cessation of services during lockdowns. Clinicians described patient fears about attending preventive services as well as their own concerns about contracting COVID while providing care. Exemplar quotes currently in the Results section are:

1. Near cessation of screening: ““People working, especially in family medicine, were distributed to the COVID clinic… And so non-essential visits including routine pap smears were put on hold” (APP, Family Medicine).”

2. Risk of contracting COVID outweighed benefits for patients: “Patients were hesitant, especially in the first year of [the] COVID pandemic, to leave their home for unnecessary reasons, including screening tests such as Pap smear” (MD, Family Medicine).

3. Risk of contracting COVID outweighed benefits for clinicians: “before we were able to be vaccinated… it felt like unnecessary risk” (APP, Women’s Health).

6c. Lines 331-333: Quantitative findings referred to in this sentence were not included in the Results section nor relevant tables. It would be informative to provide a breakdown of screens provided by speciality in Table 1.

We appreciate this suggestion and we have now created a separate table that displays cervical cancer screenings performed monthly by clinician specialty. This is now Table 2. We have also added this information into the Results section on pg. 7.

6d. Lines 335-336: The statement that APPs performed more screens than physicians has not been included in the results. It would be informative to provide a breakdown of screens provided per training in Table 1.

As indicated above, we have now created a separate table that displays cervical cancer screenings performed monthly by clinician training. This is now Table 2. We have also added this information into the Results section.

6e. Lines 367-369 These themes have already been raised earlier in the discussion (lines 339-346). Suggest merging the two relevant paragraphs.

We have made this suggested change.

7. Table 1: No details on staffing are provided in this table; title 9 rows from the end of the table should be amended.

Thank you for your attention to detail. We have amended the sub-title in this section of the table.

8a. Table 2: recommend adding zeros before the point for more clarity.

Zeros have been added to Table 2 as suggested.

8b. Table 2: a footnote listing the variables for which regression was adjusted should be listed.

At the end of the title of table 2 (now table 3), we state:

“Manual forwards selection was utilized, and the following variables were not selected for the final model (p >.10): (1) region (2) gender and (3) age.”

8c. Table 3: add 'adjusted' to 'odds ratio'.

Adjusted’ has been added to the table.